# Population Genetic Analysis Reveals Recent Demographic Expansion and Local Differentiation of Areca Palm Velarivirus 1 in Hainan Island

**DOI:** 10.3390/plants14192952

**Published:** 2025-09-23

**Authors:** Xiaoqing Niu, Zhongtian Xu, Zhaowei Lin, Qinghua Tang, Zhenguo Du, Fangluan Gao

**Affiliations:** 1Coconut Research Institute, Chinese Academy of Tropical Agricultural Science, Wenchang 571339, China; linzhaowei163@163.com (Z.L.); tchuna129@163.com (Q.T.); 2Institute of Plant Virology, Fujian Agriculture and Forestry University, Fuzhou 350002, China; xuzhongtian@nbu.edu.cn (Z.X.); duzhenguo1228@163.com (Z.D.); 3Institute of Plant Virology, Ningbo University, Ningbo 315211, China

**Keywords:** areca palm, areca palm velarivirus 1, genetic diversity, population structure, demographic expansion, natural selection

## Abstract

Areca palm velarivirus 1 (APV1), the causal agent of yellow leaf disease (YLD), poses a serious threat to the economically important areca palm industry in the Hainan Province, China, yet its evolutionary dynamics remain poorly understood. Here, we performed a large-scale molecular survey by sequencing the coat protein (CP) gene from 364 APV1-infected samples collected across major cultivation regions of Hainan. Population genetic analyses revealed extremely high haplotype diversity (*H*_d_ = 0.997) but very low nucleotide diversity (*π* = 0.017). Neutrality tests (Tajima’s *D* = −2.266; Fu’s *F*_S_ = −23.697) and a unimodal mismatch distribution supported a scenario of recent demographic expansion from a restricted ancestral pool. Evolutionary analyses indicated that the *CP* gene is subject to strong purifying selection, although eight codons exhibited episodic positive selection, suggesting ongoing viral adaptation. Furthermore, we identified three distinct genetic clusters with significant geographic structuring, indicating that viral dissemination is shaped by local factors. Together, these results reveal a recent explosive invasion of APV1 characterized by rapid island-wide expansion and emerging local differentiation. This work provides novel insights into the evolutionary trajectory of APV1 and establishes a genomic basis for improved surveillance and management of YLD.

## 1. Introduction

The areca palm (*Areca catechu* L.) is a cash crop of substantial economic importance, cultivated extensively across South and Southeast Asia, the Pacific Islands, and parts of Africa [1]. In China, the Hainan Province, which is located on an island in the South China Sea, dominates production and accounts for approximately 99% of the national output [2]. This industry plays a pivotal role in the regional economy, supporting the livelihoods of nearly five million residents and contributing over 30% of annual household income for local farmers. However, areca palm cultivation faces a severe threat from yellow leaf disease (YLD), a devastating condition first reported in India in 1914 [3]. YLD manifests as progressive leaf yellowing and reduced plant vigor, leading to significant losses in yield and quality [1]. Although its etiology has been debated for decades—with early studies implicating phytoplasmas [4]—mounting evidence now identifies areca palm velarivirus 1 (APV1) as the primary causal agent in Hainan [3,5,6].

APV1, a member of the genus *Velarivirus* (family *Closteroviridae*), possesses a positive-sense, single-stranded RNA genome containing 11 open reading frames (ORFs) [7]. Its first open reading frame, ORF1a, encodes a large multifunctional polyprotein that includes papain-like protease (P-Pro), methyltransferase (Met), and helicase (Hel) domains. ORF1b, expressed via a -1 ribosomal frameshift from ORF1a, encodes the RNA-dependent RNA polymerase (RdRp). The downstream ORFs encode a small hydrophobic protein (ORF2), a heat shock protein 70 homolog (HSP70h, ORF3), a 64 kDa polypeptide (ORF4), and a protein of unknown function (ORF5). ORF6 and ORF7 encode the major coat protein (CP) and minor coat protein (CPm), respectively, while the functions of proteins from ORFs 8–10 remain uncharacterized [8].

The causal association between APV1 and YLD is substantiated by multiple lines of compelling evidence. APV1 was first identified in symptomatic areca palms from Hainan via small RNA sequencing [3]. Subsequent RT-PCR surveys confirmed its high prevalence in diseased plants across the province, with only minimal detection in asymptomatic counterparts [5,9]. Quantitative analyses further revealed a strong positive correlation between APV1 titers and symptom severity [10]. Moreover, transmission experiments demonstrated that APV1 is efficiently vectored to healthy seedlings by mealybugs (*Pseudococcus cryptus* and *Ferrisia virgata*), which subsequently develop typical YLD symptoms [6].

Despite this etiological clarity, significant gaps persist in our understanding of APV1′s epidemiology and evolutionary dynamics. Key unanswered questions include whether the APV1 population in Hainan resulted from long-term local evolution or a recent introduction, and how its spatial spread and adaptive processes are governed. To address these questions, we conducted a comprehensive molecular survey of APV1 across the Hainan Province. By sequencing the *CP* gene from over 360 infected samples collected from major production areas, we aimed to delineate the virus’s genetic diversity, population structure, and evolutionary patterns. This work provides a crucial foundation for understanding APV1 epidemiology and developing genome-informed management strategies for YLD.

## 2. Results

### 2.1. APV1 Exhibits High Haplotype Diversity but Low Nucleotide Diversity

A total of 364 APV1 isolates were randomly collected from areca palm (*Areca catechu* L.) across 76 towns spanning 18 counties/cities in the Hainan Province between 2020 and 2022 (Appendix A). The sampling locations spanned latitudes from 18.34° N to 19.89° N and longitudes from 108.86° E to 110.58° E, representing the major cultivation areas across the island (Figure 1). Sequence analysis of the CP sequences from these isolates revealed 287 distinct haplotypes, corresponding to an exceptionally high haplotype diversity (*H*_d_ = 0.997). This indicates a rich variety of sequence variants within the population. Despite this, nucleotide diversity was remarkably low (*π* = 0.017 ± 0.0003), with the minimum pairwise identity of 95.8% (Table 1). This combination—high haplotype diversity coupled with low nucleotide diversity—is a hallmark of populations that have recently expanded from a limited ancestral pool.

To contextualize this finding, we calculated the nucleotide diversity for two other velariviruses, grapevine leafroll-associated virus 7 (GLV7) and little cherry virus 1 (LCV1). Both viruses exhibited significantly higher diversity (*π* = 0.067 ± 0.008 and 0.206 ± 0.007, respectively), indicating that the low nucleotide diversity observed in APV1 is an unusual feature for this genus and likely reflects its unique evolutionary history in Hainan (Table 1).

### 2.2. Neutrality Tests and Mismatch Distribution Support APV1 Population Expansion

Neutrality tests yielded significantly negative values for both Tajima’s *D* (−2.266, *p* < 0.01) and Fu’s *F*_S_ (−23.697, *p* < 0.01) (Table 1). These results are consistent with an excess of low-frequency mutations, a pattern indicative of either population growth or a recent selective sweep. To distinguish between these possibilities, we performed a mismatch distribution analysis. The observed distribution of pairwise nucleotide differences was distinctly unimodal and smooth, fitting well with the expectations of a spatial expansion model (*SSD* = 0.002, *p* > 0.05; *H*_rag_ = 0.005, *p* > 0.05) (Figure 2).

Collectively, while negative neutrality statistics alone can be ambiguous, the unimodal mismatch distribution strongly suggests that demographic expansion is the primary driver shaping the genetic structure of the APV1 population in Hainan.

### 2.3. The APV1 CP Gene Is Under Strong Purifying Selection with Episodic Adaptation

We evaluated selective pressures on the *CP* gene by calculating the ratio of nonsynonymous to synonymous substitution rates (*d*N/*d*S). The overall ratio was substantially less than one, indicating that strong purifying selection is the dominant evolutionary force acting on this gene (Figure 3). This finding is consistent with the critical functional constraints on the CP, which is essential for virion assembly, vector transmission, and host interactions.

To detect potential episodic adaptation, we performed a site-specific selection analysis using CODEML4.7. Likelihood ratio tests revealed that models incorporating positive selection (M2a and M8) provided a significantly better fit to the data than their neutral counterparts (M1a and M7) (*p* < 0.001). This result indicates the presence of specific codons evolving under positive selection. Under model M8, six codons (positions 9, 53, 87, 121, 149, and 188) were identified as positively selected, with the strongest evidence (posterior probability > 0.95) for these sites (Figure 3, Table 2). Taken together, these results suggest that while the *CP* gene is overwhelmingly constrained by purifying selection, a few specific sites may undergo positive selection, potentially as a result of adaptive pressures related to host interaction.

### 2.4. APV1 Populations Exhibit Significant Geographic Structuring

To evaluate the spatial distribution of genetic variation, we performed a Discriminant Analysis of Principal Components (DAPC), which resolved three well-supported genetic clusters with distinct geographic distributions (Figure 4). Cluster 1 was widespread across the island, suggesting it represents a broadly disseminated lineage. In contrast, Cluster 2 was more localized, primarily restricted to Ledong, Lingshui, Ding’an, and Qionghai. Cluster 3 was highly geographically constrained, found exclusively in Chengmai (Figure 1 and Figure 4).

This population subdivision was further supported by fixation index (*F*_ST_) analysis. Although pairwise *F*_ST_ values among clusters were low in magnitude—a consequence of the overall limited nucleotide diversity of APV1—they were nevertheless statistically significant, indicating restricted gene flow among these regional populations (Table 3).

We further tested this phylogeographic structure using association index (*AI*), parsimony score (*PS*), and monophyletic clade (*MC*) statistics (Table 4). Both *AI* and *PS* values were significantly lower than expected under a null hypothesis of no geographic association (*p* < 0.001), confirming a strong correlation between viral phylogeny and geographic origin. Moreover, the MC statistic for each of the three clusters was significant (*p* < 0.01), indicating that they represent distinct, geographically cohesive lineages that are more phylogenetically related than expected by chance.

In summary, these complementary analyses provide robust evidence that the APV1 population in Hainan is not panmictic but is instead characterized by significant geographic differentiation. This structuring is likely shaped by a combination of localized vector-driven transmission, limited vector dispersal, and human-mediated movement of planting materials, which collectively maintain distinct viral subpopulations.

## 3. Discussion

In this study, we generated a high-resolution dataset by sequencing 364 APV1 *CP* genes from samples collected across Hainan Island (Figure 1, Appendix A). This represents a substantial increase in sampling density compared with previous studies, which, despite relying on whole-genome sequencing, included only 20 isolates [8]. The expanded dataset provides a more comprehensive view of APV1 genetic diversity, population structure, and evolutionary dynamics, enabling more robust inferences about its epidemiology and adaptive patterns.

A notable observation from our analysis concerns the origin and demographic history of APV1 in Hainan. The combined patterns of high haplotype diversity, low nucleotide diversity, negative neutrality statistics (Table 1), and a unimodal mismatch distribution (Figure 2) are not consistent with long-term, stable viral endemicity. Instead, these features collectively point toward a scenario in which the APV1 population may have undergone a relatively recent demographic expansion from a limited ancestral pool, likely shaped by a founder effect in which a small number of founding genomes established local populations that subsequently expanded. Comparable demographic signatures have been reported in geographically isolated virus populations, where introduction events followed by rapid growth produce elevated haplotype richness but low overall sequence divergence [11,12,13,14].

Despite the limited genetic diversity expected under a founder effect and the relatively isolated nature of host populations in Hainan, APV1 displays unexpectedly widespread occurrence, a pattern that may in part reflect the wide distribution and ecological plasticity of its mealybug vectors. Nevertheless, this genetic signature of recent expansion presents an apparent paradox when considered alongside historical reports of YLD in Hainan dating back to 1985 [15]. Several non-exclusive hypotheses could reconcile this discrepancy. First, the now-dominant APV1 lineage may have circulated at a very low prevalence for decades before a recent surge in incidence, potentially triggered by intensified cultivation or shifts in vector ecology. Second, this lineage could have competitively displaced older, more divergent APV1 strains that are no longer detectable. Third, YLD symptoms are highly complex and may result from multiple biotic and abiotic factors [1]. Early YLD-like symptoms might have been caused by other pathogens, with APV1 emerging later as the principal etiological agent. Although further data are needed to distinguish among these scenarios, our results are consistent with the interpretation that most present-day APV1 infections derive from a relatively recent common ancestor, underlining the dynamic nature of viral emergence in agroecosystems.

Our analysis of the evolutionary forces shaping the *CP* gene provides further mechanistic insight. The gene is overwhelmingly constrained by strong purifying selection (Figure 3), reflecting stringent functional requirements for virion assembly, vector transmission, and host interaction. However, superimposed on this highly conserved background, our site-specific analyses identified discrete codons under positive selection (Table 2, Figure 3). This dual-mode evolution—wherein the structural integrity of the protein is rigorously maintained while specific sites undergo adaptive change—suggests a sophisticated viral strategy. These adaptive “hotspots” may allow APV1 to fine-tune its interface with host defenses or vector components, facilitating persistence and spread without compromising core functions.

The significant geographic structuring of APV1 (Figure 4) adds a crucial layer of nuance to the expansion narrative. Although the overall genetic diversity is low, the three distinct clusters reveal that the virus’s spread has not been a simple, uniform wave (Figure 4). Instead, this nascent differentiation points to a dynamic interplay between island-wide expansion and local-level drivers, such as restricted vector dispersal and regional patterns of human-mediated transport of planting materials. This suggests that even as APV1 rapidly colonizes Hainan, it is already beginning to differentiate in response to local selective pressures, a process that may presage the emergence of distinct regional lineages in the future.

From an applied perspective, our findings have critical implications for disease management. The genetic signature of rapid expansion highlights APV1′s high invasive potential within intensively managed agroecosystems. Furthermore, the evidence for positive selection, though limited to a few sites, signals that the virus possesses the capacity to adapt, potentially leading to the evolution of altered virulence, transmissibility, or host range. These realities underscore the urgent need for proactive genomic surveillance to monitor the emergence of new variants and track adaptive changes in real time. Integrating this molecular intelligence with on-farm strategies—such as screening planting material, developing resistant cultivars, and implementing targeted vector control—is essential for devising robust and sustainable solutions to mitigate the devastating impact of YLD.

A primary limitation of this study is its focus on a single gene. While the *CP* gene is highly informative for population genetic analyses, it represents only a fraction of the viral genome. Consequently, our analysis cannot capture evolutionary dynamics in other functionally important regions, nor can it definitively detect recombination events. Future research employing large-scale, whole-genome sequencing will be crucial for building a more complete picture of APV1′s evolution and adaptation. Nevertheless, the depth of sampling in our study provides a foundational understanding of APV1 population dynamics that can guide these future genomic efforts.

## 4. Materials and Methods

### 4.1. Sample Collection and Processing

APV1-infected areca palm (*Areca catechu* L.) leaf samples were collected between 2020 and 2022 from multiple cultivation areas across the Hainan Province (Appendix A). Symptomatic leaves were excised using sterile scissors and gloves to prevent cross-contamination. Samples were immediately placed in labeled plastic bags, transported on ice to the laboratory, and stored at −80 °C until processing.

### 4.2. Virus Isolates and Genetic Diversity Analysis

To obtain the APV1 CP sequences, total RNAs was extracted using an RNAprep Pure Plant Plus Kit (Polysaccharides and Polyphenolics-rich), and cDNA was synthesized by a FastKing RT Kit (both from TIANGEN, Beijing, China), according to the manufacturer’s instructions. RT PCR amplifications were performed in a total volume of 50 μL composed of 25 μL × PCR *taq* mix, 2.5 μL of forward primer (10 μmol/L, APV1CP_F1: 5′-CCACTCTTCTGGTAGTATCAAGG-3′), 2.5 μL of reverse primer (10 μmol/L, APV1CP_R1: 5′-CAGAAGCATAAGATTGTGACATTTTTACCG-3′), 15 μL of ddH_2_O, and 5.0 μL of template cDNA. The thermal profile included an initial denaturation at 94 °C for 3 min, followed by 34 cycles of 94 °C for 30 s, 55 °C for 30 s, and 72 °C for 1 min, with a 10 min extension at 72 °C. Following the completion of PCR, the amplified product was recovered and ligated into the cloning vector PMD-18T. Culture solution was then carried out to identify successful results, after which the solution was sequenced in both directions by Sangon Biological Co., Ltd. (Shanghai, China).

A codon-based alignment of the APV1 CP sequence was performed using the MAFFT algorithm [16], as recommended in PhyloSuite 1.31 [17]. To assess APV1 populations’ genetic diversity, haplotype diversity and nucleotide diversity were calculated using DnaSP 6.12.03 [18]. For comparative purposes, the same analysis was performed on the *CP* genes sequences of two other velariviruses, GLV7 and LCV1 (Appendix A).

### 4.3. Demographic History of the APV1 Population

All mismatch distribution analyses and parameter calculations were performed using Arlequin 3.5.2.2 [19]. To test recent demographic expansion, we calculated Tajima’s *D* [20] and Fu’s *F*_S_ [21] statistics, where significantly negative values suggested potential population expansion or selective sweeps. The expansion model was further evaluated through Harpending’s raggedness index (*H*_rag_) and sum of squared deviations (*SSD*), with their corresponding *p*-values [22]. Non-significant *p*-values (*p* > 0.05) for both *H*_rag_ and *SSD* supported the population expansion hypothesis, while unimodal mismatch distributions provided additional evidence for this demographic scenario.

### 4.4. Detecting Natural Selection

To quantify patterns of natural selection acting on APV1 CP, the ratio of non-synonymous (*d*N) to synonymous (*d*S) substitution rates (*ω* = *d*N/*d*S) was calculated using the CODEML algorithm implemented in EasyCodeML 1.41 [23,24]. This widely used metric for evaluating selective pressures on protein-coding genes was obtained through site-model comparisons of four nested model pairs: M0 (one ratio) versus M3 (discrete), M1a (neutral) versus M2a (selection), M7 (beta) versus M8 (beta and ω > 1), and M8 versus M8a. Positively selected amino acid sites in APV1 CP were identified through Bayes Empirical Bayes analysis when likelihood-ratio tests showed statistical significance (*p* < 0.05) [25].

### 4.5. Testing Population Differentiation

Genetic differentiation among populations was assessed using *K*_ST_, *Z*, and S_nn_ statistics in DnaSP 6.12.03 [18], with significance tested through 1000 permutations of the original dataset. In addition, pairwise *F*_ST_ values were also computed in Arlequin 3.5 [19] and interpreted as follows: 0.05–0.15 indicated moderate differentiation, 0.15–0.25 represented high differentiation, and values above 0.25 suggested very strong genetic divergence [26]. These *F*_ST_ estimates also provided insight into gene flow patterns as well, with values below 0.33 indicating substantial gene flow and higher values suggesting more restricted genetic exchange between populations [27].

To further investigate the genetic structure of APV1 populations, discriminant analysis of principal components (DAPC) was performed using the adegenet 2.1.11 R package to analyze predefined groups [28]. In contrast to methods that assume panmixia, DAPC partitions genetic variance into between-group and within-group components, effectively maximizing discrimination among populations.

### 4.6. Phylogeny-Geography Correlation of APV1

To evaluate geographic influences on APV1 CP diversification, we conducted phylogeny-trait association tests using BaTS 2.0 [29]. The phylogenetic signal was quantified through three metrics: association index (*AI*), parsimony score (*PS*), and maximum monophyletic clade (*MC*). Statistical significance (*p* < 0.05) was determined by comparing observed values against 1000 randomized location permutations, with significant results indicating strong phylogeny-geography correlations.

## Figures and Tables

**Figure 1 plants-14-02952-f001:**
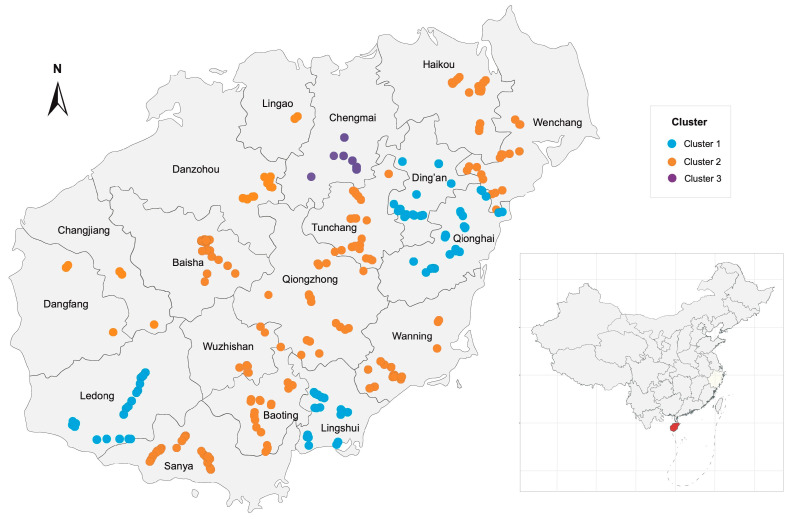
Geographical distribution of APV1 sampling sites in Hainan. Circles represent sampling locations, and colors indicate the genetic clusters of the isolates (as defined by subsequent discriminant analysis of principal components).

**Figure 2 plants-14-02952-f002:**
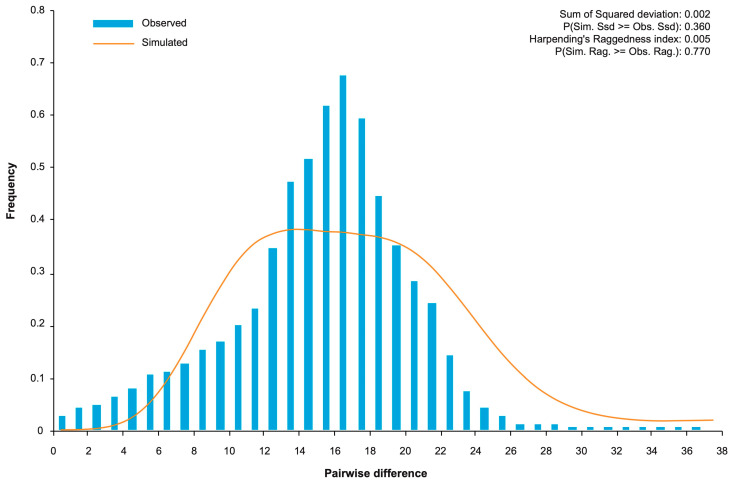
Demographic history dynamics of the APV1 population inferred from mismatch distribution analysis. The histogram shows the observed pairwise genetic differences, while the orange line represents the expected distribution under a population expansion model.

**Figure 3 plants-14-02952-f003:**
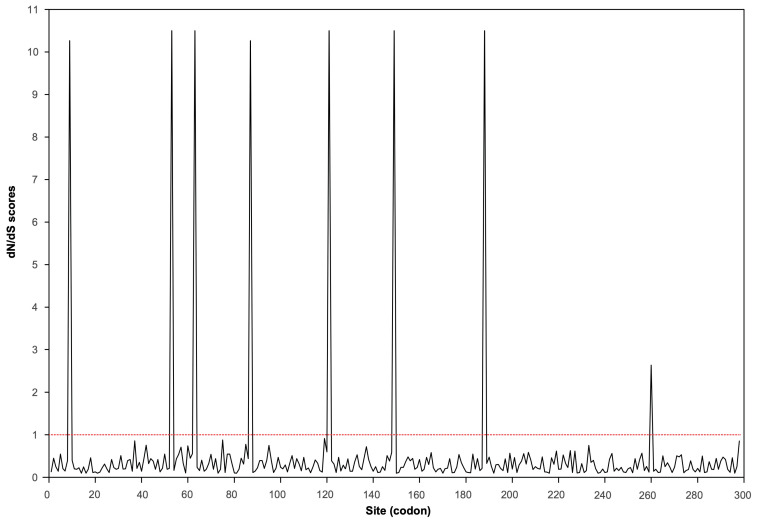
Sliding window analysis of *d*N/*d*S ratios across the APV1 *CP* gene. The red dotted line denotes the neutral expectation (*d*N/*d*S = 1).

**Figure 4 plants-14-02952-f004:**
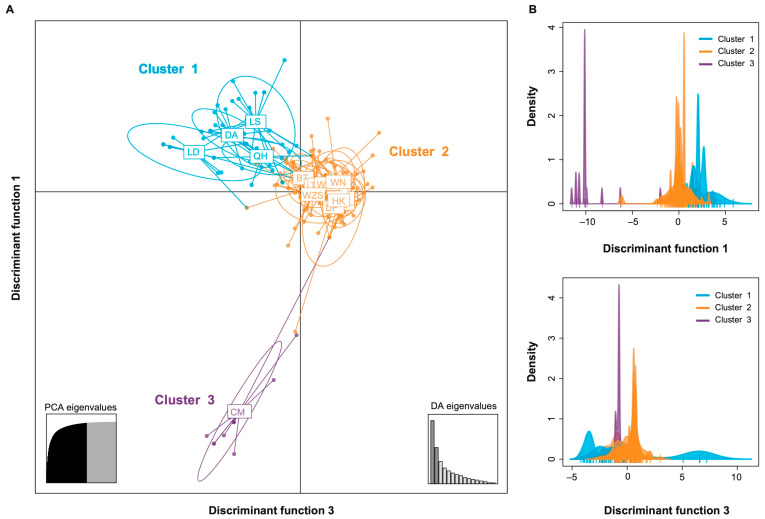
Discriminant analysis of principal components (DAPC) of predefined APV1 geographic populations. (**A**) Individual isolates are represented as dots, with inertia ellipses encompassing the majority of each group. The inset panel displays eigenvalues for the principal component analysis (PCA) and discriminant analysis (DA). (**B**) Cluster separation using the first and second discriminant functions. Abbreviations correspond to the following counties/cities: BS, Baisha; BT, Baoting; CJ, Changjiang; CM, Chengmai; DA, Ding’an; DF, Dongfang; DZ, Danzhou; HK, Haikou; LD, Ledong; LG, Lingao; LS, Lingshui; QH, Qionghai; QZ, Qiongzhong; SY, Sanya; TC, Tunchang; WC, Wenchang; WN, Wanning; WZS, Wuzhishan.

**Table 1 plants-14-02952-t001:** Estimation of genetic diversity parameters and neutrality tests for the populations of APV1 and two other velariviruses.

Virus	*n*	*h*	Haplotype Diversity	Nucleotide Diversity	Tajima’s *D*	Fu’s *F*_S_
APV1	364	287	0.997 ± 0.001	0.017 ± 0.0003	−2.266 ***	−23.697 **
GLV7	17	13	0.949 ± 0.008	0.067 ± 0.008	−0.818 ^ns^	3.442 ^ns^
LCV1	57	50	0.999 ± 0.004	0.206 ± 0.007	2.179 ^ns^	−2.915 ^ns^

*n*, sample size; *h*, number of haplotypes; Significance thresholds: **, 0.001 < *p* < 0.01; ***, *p* < 0.001; ns, not significant.

**Table 2 plants-14-02952-t002:** Positively selected sites detected in the APV1 CP.

Model	*np*	*ln* L	Model Compared	LRT *p*-Value	Positively Selected Sites
M1a	64	−5473.213			Not allowed
M2a	66	−5473.108	M1a vs. M2a	< 0.001	[]
M7	64	−5476.803			Not allowed
M8	66	−5472.930	M7 vs. M8	< 0.001	9 *, 53 **, 63 **, 87 *, 121 **, 149 **, 188 **
M8a	65	−5473.183	M8a vs. M8	< 0.001	Not allowed

*np*, number of parameters; LRT, likelihood ratio test; [], no data available. Significance thresholds: *, Posterior probability (PP) >0.095; **: PP > 0.99.

**Table 3 plants-14-02952-t003:** Population differentiation tests for the APV1 population.

Population	*K* _ST_	*Z*	*S* _nn_	*F* _ST_
Cluster 1 vs. Cluster 2	0.021 ***	28821.739 ***	0.877 ***	0.055 ***
Cluster 1 vs. Cluster 3	0.057 ***	2455.444 ***	0.946 ***	0.153 ***
Cluster 2 vs. Cluster 3	0.018 ***	19545.859 ***	0.957 ***	0.117 ***

Significance thresholds: ***, *p* < 0.001.

**Table 4 plants-14-02952-t004:** Phylogeny-trait association tests of APV1.

Statistic	Observed Mean (95% CI)	Null Mean (95% CI)	*p*-Value
*AI*	4.964 (3.889–6.081)	17.694 (16.640–18.678)	<0.001
*PS*	40.706 (37.000–43.000)	93.567 (91.058–96.190)	<0.001
*MC* (Cluster 1)	17.946 (18.000–19.000)	2.504 (2.220–3.019)	<0.010 **
*MC* (Cluster 2)	39.070 (39.00–40.000)	9.009 (7.365–11.975)	<0.010 **
*MC* (Cluster 3)	5.680 (3.000–10.000)	1.326 (1.026–1.995)	<0.010 **

*AI*, association index; *PS*, parsimony score; *MC*, maximum monophyletic clade; 95% CI, 95% confidence interval. Significance thresholds: **, 0.001 < *p* < 0.01.

## Data Availability

All data used in this study are publicly available on NCBI. The *CP* gene sequence obtained in this study has been deposited in GenBank under the accession numbers PX243804-PX244167.

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
