# Peer review of "Population Genetic Analysis Reveals Recent Demographic Expansion and Local Differentiation of Areca Palm Velarivirus 1 in Hainan Island"

_plants, 2025, doi:10.3390/plants14192952_

Round 1

Reviewer 1 Report

Comments and Suggestions for Authors

This paper provides a global profiles of areca palm velarivirus 1 (APV1) causing yellow leaf disease on areca palm in Hainan island in China. It is of scientific importance in illuminating the evolution of APV1 and providing genomic data for improved strategies to control APV1. The paper is well presented, expect some minors should be improved.

  1. Hainan Island faces the Chinese mainland across the sea. There should be a short introduction of the geographic location of Hainan Island. It should be made clear that Hainan Island is actually Hainan Province; otherwise, non-Chinese people might confuse.
  2. Figure 4. Several abbreviations are presented in this figure. Give the full names of these abbreviations. It is necessary to ensure the charts are self-explanatory.
  3. Line 115, 128……, CP, here refers to gene, should be italic.
  4. Line 248, revise 72° into 72°C.

Author Response

COMMENTS:

This paper provides a global profile of areca palm velarivirus 1 (APV1) causing yellow leaf disease on areca palm in Hainan island in China. It is of scientific importance in illuminating the evolution of APV1 and providing genomic data for improved strategies to control APV1. The paper is well presented, expect some minors should be improved.

RESPONSE: We thank the reviewer for these positive comments.

COMMENTS:

Hainan Island faces the Chinese mainland across the sea. There should be a short introduction of the geographic location of Hainan Island. It should be made clear that Hainan Island is actually Hainan Province; otherwise, non-Chinese people might confuse.

RESPONSE: We thank the reviewer for pointing this out. As suggested, we have now incorporated a concise geographic descriptor into the manuscript to explicitly clarify that Hainan is an island province, thereby preventing potential confusion for international readers.

COMMENTS:

Figure 4. Several abbreviations are presented in this figure. Give the full names of these abbreviations. It is necessary to ensure the charts are self-explanatory.

RESPONSE: We have now included the full names of all geographic abbreviations in the revised figure legend to ensure clarity and self-explanatory presentation.

COMMENTS:

Line 115, 128……, CP, here refers to gene, should be italic.

RESPONSE: We have ensured that all instances of "CP" referring to the gene are now italicized throughout the manuscript.

COMMENTS:

Line 248, revise 72° into 72°C.

RESPONSE: Change as suggested.

Reviewer 2 Report

Comments and Suggestions for Authors

The authors, Xiaoqing Niu and co-workers, presented a manuscript of a communication entitled, " Population genetic analysis reveals recent demographic expansion and local differentiation of areca palm velarivirus 1 in Hainan Island ".

The authors analyzed a large number of samples of areca palm trees infected with areca palm velarivirus (APV1) in the island province of Hainan, China, and sequenced the gene for the virus coat protein. They then performed a very thorough population analysis of the virus. They found that the studied population has very high haplotype diversity and low nucleotide diversity. The authors conclude that the virus was introduced into the analyzed area relatively recently.

The paper is well written. Robust population analysis tools were chosen, and the methods and procedures used are well described. The results are presented in considerable detail and compared with previous work by other authors.

The contribution of the work lies in the thorough bioinformatic analysis of the virus population, which is important for agriculture.

To improve the quality of the article, I recommend:

Discussing the distribution and ecology of the virus vector in Hainan.

Expanding the discussion to include other published work on this virus.

Expanding the discussion to include similar analyses focused on isolated virus populations.

Author Response

COMMENTS:

The authors, Xiaoqing Niu and co-workers, presented a manuscript of a communication entitled, " Population genetic analysis reveals recent demographic expansion and local differentiation of areca palm velarivirus 1 in Hainan Island ".

The authors analyzed a large number of samples of areca palm trees infected with areca palm velarivirus (APV1) in the island province of Hainan, China, and sequenced the gene for the virus coat protein. They then performed a very thorough population analysis of the virus. They found that the studied population has very high haplotype diversity and low nucleotide diversity. The authors conclude that the virus was introduced into the analyzed area relatively recently.

The paper is well written. Robust population analysis tools were chosen, and the methods and procedures used are well described. The results are presented in considerable detail and compared with previous work by other authors.

The contribution of the work lies in the thorough bioinformatic analysis of the virus population, which is important for agriculture.

RESPONSE: We thank the reviewer for these positive comments.

COMMENTS:

To improve the quality of the article, I recommend:

Discussing the distribution and ecology of the virus vector in Hainan.

RESPONSE:

RESPONSE: We appreciate the reviewer’s suggestion to discuss the distribution and ecology of the virus vector in Hainan. In our original discussion, we had already mentioned the role of mealybugs (Pseudococcus cryptus and Ferrisia virgata) as efficient vectors of APV1. Following the reviewer’s advice, we have now added an additional sentence highlighting the wide distribution and ecological plasticity of these mealybugs in Hainan, to better contextualize their contribution to APV1 epidemiology.

COMMENTS:

Expanding the discussion to include other published work on this virus.

RESPONSE: We thank the reviewer for this valuable suggestion. In the revised discussion, we have expanded the contextualization of our results with reference to previously published studies on APV1. Nevertheless, research on APV1 remains limited, particularly with respect to population genetics. This scarcity underscores the significance of our study, which provides a high-resolution dataset and a detailed analysis of APV1 genetic diversity, population structure, and evolutionary dynamics, thereby contributing novel insights to the field.

COMMENTS:

Expanding the discussion to include similar analyses focused on isolated virus populations.

RESPONSE: We thank the reviewer for this valuable suggestion. In the revised discussion, we have added a sentence highlighting that the genetic signature of recent expansion in APV1 is consistent with patterns observed in other studies of isolated virus populations.

Reviewer 3 Report

Comments and Suggestions for Authors

In the manuscript submitted to me for review entitled "Population genetic analysis reveals recent demographic expansion and local differentiation of areca palm velarivirus 1 in Hainan Island“ the authors present a study in which they trace the evolutionary dynamics of the coat protein (CP) gene of areca palm velarivirus 1 (APV1) from 364 APV1-infected areca palm samples collected in the main growing areas of Hainan.

My remarks and recommendations to the authors are:

  1. Authors can take advantage of this stage of manuscript processing to include more keywords that will make the future article more easily discoverable by readers.

The Materials and Methods section is not well presented. I have a few recommendations for it.

  1. It is good to separate the individual methods into separate subsections so that the reader can better familiarize himself with the methodologies used.
  2. The abstract states that 364 samples were studied. In Materials and Methods section nothing is mentioned about these samples. It is good to add a subsection (and the first one, before the other methods are described), in which certain information is indicated. For example, at the beginning of section 2.1. information is presented about where the samples were collected from. It would be good to add at least one sentence in Materials and Methods section with this information, or to move the one from section 2.1. here. It is also good to indicate information such as: is there a specific period of the year in which the samples were collected; how the samples were collected; the way in which they were prepared in a form suitable for laboratory testing; the method and conditions of storage.
  3. In Figure 2, the inscriptions are small and difficult to read. It would be good for readers if the font size both on the ordinate axes and inside the figure were increased. There is enough space in my opinion and this will not interfere with the presentation of the results. The same remark applies to Figure 4 and the ordinate axes of Figure 3.
  4. In Figure 4, it is good to label the three subfigures separately (for example, A, B and C) and that each subfigure should indicate under the figure what information it presents.

Author Response

COMMENTS:

In the manuscript submitted to me for review entitled "Population genetic analysis reveals recent demographic expansion and local differentiation of areca palm velarivirus 1 in Hainan Island ” the authors present a study in which they trace the evolutionary dynamics of the coat protein (CP) gene of areca palm velarivirus 1 (APV1) from 364 APV1-infected areca palm samples collected in the main growing areas of Hainan.

My remarks and recommendations to the authors are:

  1. Authors can take advantage of this stage of manuscript processing to include more keywords that will make the future article more easily discoverable by readers.

RESPONSE: Thank you for the suggestion. We have appended two additional keywords in the revised manuscript to improve the article's discoverability.

The Materials and Methods section is not well presented. I have a few recommendations for it.

COMMENTS:

  1. It is good to separate the individual methods into separate subsections so that the reader can better familiarize himself with the methodologies used.

RESPONSE: Thank the reviewer for this helpful suggestion. As suggested by the reviewer, we have now restructured the Methods section into separate subsections, each with a descriptive title, to improve clarity and help readers navigate the methodologies used.

COMMENTS:

  1. The abstract states that 364 samples were studied. In Materials and Methods section nothing is mentioned about these samples. It is good to add a subsection (and the first one, before the other methods are described), in which certain information is indicated. For example, at the beginning of section 2.1. information is presented about where the samples were collected from. It would be good to add at least one sentence in Materials and Methods section with this information, or to move the one from section 2.1. here. It is also good to indicate information such as: is there a specific period of the year in which the samples were collected; how the samples were collected; the way in which they were prepared in a form suitable for laboratory testing; the method and conditions of storage.

RESPONSE: We thank the reviewer for this suggestion. While the sampling locations and time frame were described in section 2.1 of the Results, we have now added a dedicated subsection in Materials and Methods that presents this information in full, including the period of sample collection, the procedures used to collect the samples, the preparation of samples for laboratory analysis, and storage conditions prior to testing. This ensures clarity and reproducibility for readers.

COMMENTS:

  1. In Figure 2, the inscriptions are small and difficult to read. It would be good for readers if the font size both on the ordinate axes and inside the figure were increased. There is enough space in my opinion and this will not interfere with the presentation of the results. The same remark applies to Figure 4 and the ordinate axes of Figure 3.

RESPONSE: We thank the reviewer for this helpful suggestion. The font sizes on the ordinate axes and within the panels of Figures 2, 3, and 4 have now been increased to improve readability.

COMMENTS:

  1. In Figure 4, it is good to label the three subfigures separately (for example, A, B and C) and that each subfigure should indicate under the figure what information it presents.

RESPONSE: Thank the reviewer for the valuable comment regarding the organization of Figure 4. As suggested, we have reorganized the subfigures into two panels: (A) and (B). The first subfigure is now labeled (A), while the latter two related subfigures have been combined under panel (B), which illustrates cluster separation using the first and second discriminant functions. This adjustment improves clarity and better aligns with the presentation of results.

Round 2

Reviewer 3 Report

Comments and Suggestions for Authors

The authors of the manuscript "Population genetic analysis reveals recent demographic expansion and local differentiation of areca palm velarivirus 1 in Hainan Island" have answered all my questions. The authors have made all the corrections and additions to the text suggested by me. I have no additional questions or comments for the authors.